# Efficacy of Anti-Cancer Immune Responses Elicited Using Tumor-Targeted IL-2 Cytokine and Its Derivatives in Combined Preclinical Therapies

**DOI:** 10.3390/vaccines13010069

**Published:** 2025-01-13

**Authors:** Sahar Balkhi, Giorgia Bilato, Andrea De Lerma Barbaro, Paola Orecchia, Alessandro Poggi, Lorenzo Mortara

**Affiliations:** 1Laboratory of Immunology and General Pathology, Department of Biotechnology and Life Sciences, University of Insubria, 21100 Varese, Italy; sahar.balkhi@uninsubria.it (S.B.); gbilato@uninsubria.it (G.B.); lorenzo.mortara@uninsubria.it (L.M.); 2Unit of Molecular Pathology, Biochemistry and Immunology, IRCCS MultiMedica, 20123 Milan, Italy; 3Laboratory of Comparative Physiopathology, Department of Biotechnology and Life Sciences, University of Insubria, 20145 Varese, Italy; a.delermabarbaro@uninsubria.it; 4Pathology and Experimental Immunology Operative Unit, IRCCS Ospedale Policlinico San Martino, 16132 Genova, Italy; paola.orecchia@hsanmartino.it; 5SSD Oncologia Molecolare e Angiogenesi, IRCCS Ospedale Policlinico San Martino, 16132 Genova, Italy

**Keywords:** IL-2, immunotoxins, immunocytokines

## Abstract

Effective cancer therapies must address the tumor microenvironment (TME), a complex network of tumor cells and stromal components, including endothelial, immune, and mesenchymal cells. Durable outcomes require targeting both tumor cells and the TME while minimizing systemic toxicity. Interleukin-2 (IL-2)-based therapies have shown efficacy in cancers such as metastatic melanoma and renal cell carcinoma but are limited by severe side effects. Innovative IL-2-based immunotherapeutic approaches include immunotoxins, such as antibody–drug conjugates, immunocytokines, and antibody–cytokine fusion proteins that enhance tumor-specific delivery. These strategies activate cytotoxic CD8^+^ T lymphocytes and natural killer (NK) cells, eliciting a potent Th1-mediated anti-tumor response. Modified IL-2 variants with reduced Treg cell activity further improve specificity and reduce immunosuppression. Additionally, IL-2 conjugates with peptides or anti-angiogenic agents offer improved therapeutic profiles. Combining IL-2-based therapies with immune checkpoint inhibitors (ICIs), anti-angiogenic agents, or radiotherapy has demonstrated synergistic potential. Preclinical and clinical studies highlight reduced toxicity and enhanced anti-tumor efficacy, overcoming TME-driven immune suppression. These approaches mitigate the limitations of high-dose soluble IL-2 therapy, promoting immune activation and minimizing adverse effects. This review critically explores advances in IL-2-based therapies, focusing on immunotoxins, immunocytokines, and IL-2 derivatives. Emphasis is placed on their role in combination strategies, showcasing their potential to target the TME and improve clinical outcomes effectively. Also, the use of IL-2 immunocytokines in “in situ” vaccination to relieve the immunosuppression of the TME is discussed.

## 1. Introduction

Cancer therapies have been revolutionized by the use of immune checkpoint inhibitors (ICIs) both as treatments alone and in combination with other approaches [1,2,3,4,5,6,7,8,9,10]. Initially administered to patients with metastatic melanoma and metastatic renal cell carcinoma (RCC), these therapies have since been extended to other cancer types, achieving excellent results and, in some cases, definitive cures [1,4,8]. However, it became apparent that a significant portion of the patients did not respond to ICIs [11,12,13,14]. Among the various potential reasons for this, two critical factors must be considered: the inhibitory role of the TME on immune cells and its proactive role in promoting carcinogenesis and tumor progression [15,16,17,18,19,20].

Of note, in 2011, Hanahan and Weinberg [21] redefined and highlighted several new key hallmarks of cancer, including enabling characteristics that were initially postulated in 2000 [22]. Of particular importance are three hallmarks that play fundamental roles in tumor dynamics: “evading immune destruction”, “tumor-promoting inflammation”, and “immune orchestration of angiogenesis”. These characteristics underscore the essential and dual role of the immune system in neoplastic diseases.

Immune cells, particularly innate immune cells, such as macrophages and natural killer (NK) cells, can exhibit potent anti-tumor activities, including phagocytosis and cytotoxic functions [23,24,25]. However, at the same time, macrophages and NK cells upon interactions with the tumor and TME, can become protumor effectors with low cytotoxic capacities and display immunosuppressive and pro-angiogenic behaviors [26,27,28].

In this context, Hanahan recently suggested new emerging characteristics of tumor cells, “unlocking phenotypic plasticity”, “non-mutational epigenetic reprogramming”, “polymorphic microbiomes”, and “senescent cells”, making the study of malignant tumor growth even more complex and intricate [29]. However, increasing experimental data have begun to clarify the protumor or anti-tumor roles played by different types of immune or stromal cells, particularly in promoting or suppressing tumor growth, reinforcing or inhibiting metastasis and contributing to therapy.

Hanahan defined “unlocking phenotypic plasticity” as a potential new hallmark of cancer, referring specifically to malignant cells. However, from the point of view of the immune cells in the TME, it appears that these cells can also exhibit high plasticity in both their phenotype and function. Indeed, as a result of tumor signaling and the release of soluble factors or extracellular vesicles, immune cells can become highly plastic, altering their phenotype and function. These alterations, induced by interactions with cancer, stromal and endothelial cells, as well as cytokines/chemokines in the TME, can lead to the impairment of cytotoxic activities, the development of immunosuppressive features, and, in many cases, the acquisition of pro-angiogenic activities.

Among these factors, cytokines play a crucial role in shaping the cancer-induced immune cycle by prompting various processes, such as tumor antigen expression, antigen presentation, priming and activation of immune cells, recruitment of effector immune cells to the tumor site, and ultimately facilitating cancer cells elimination within the TME [30]. On the other hand, proinflammatory cytokines might have both protumor or anti-tumor effects. For IL-2, several lines of evidence have shown that this cytokine serves as a crucial regulator for the activation of helper/regulatory T cells, cytotoxic T cells, and NK cells, influencing their proliferation, differentiation, and contributing to both pro- and anti-inflammatory immune responses. Some specific cytokines are identified as promising cancer immunotherapies. They function by adjusting the immune response against cancer cells and by directly displaying anti-cancer effects, including inhibiting proliferation and promoting apoptosis. Cytokines have a substantial track record as anti-cancer treatments, starting from the 1970s, when interferon (IFN)-α and IL-2 were the first cytokines utilized in cancer therapy [30]. Among the various types of immunotherapies, cytokine therapy shows high promise in cancer treatment. IL-2 treatment has shown significant efficacy in metastatic melanoma, RCC, and advanced non-Hodgkin’s lymphoma. However, following these early advancements, the clinical translation of these molecules has been significantly restricted due to their broad and diverse biological effects across various cell types. These characteristics, combined with suboptimal pharmacokinetics, such as short half-lives, have impeded the effective systemic administration of cytokines, primarily due to severe dose-limiting toxicities [31]. In fact, severe adverse reactions, such as vascular leakage syndrome, underscore the need for the careful administration under the supervision of an oncologist in a hospital setting. To address these challenges, novel engineering strategies been developed to broaden the therapeutic range, extend the duration of pharmacokinetic effects, improve tumor-specific targeting, and minimize adverse effects, thereby enhancing the overall efficacy of the therapy [31]. Herein, we focus on discussing the impact and relevance of anti-cancer immune effectors induced by IL-2 immunotoxins (IT), IL-2 immunocytokines, and IL-2 derivatives or variants in different combined therapies. Also, the use of these therapeutic tools in “in situ” tumoral vaccination is discussed.

## 2. Main Features of IL-2 and Tumor-Specific Targeting of IL-2

Since its discovery in 1976 and the subsequent understanding of its vital role as a lymphocyte T-cell growth factor, the cytokine IL-2 was approved in the 1990s for use in patients with advanced-stage cancer, including RCC and later for metastatic melanoma. IL-2 is a 15.5 kDa glycoprotein involved in immune response [32,33,34]. IL-2 exerts its function by interacting with IL-2 receptors (IL-2Rs) [34,35,36]. The IL-2R is comprised of three different subunits: IL2Rα (CD25), IL2-Rβ (CD122), and IL-2Rγ (CD132). The strongest binding affinity is observed when IL-2 binds to the trimeric IL-2Rαβγ complex, though it can also bind to the dimeric IL-2Rβγ [37,38]. This implies that the lymphocytes expressing the trimeric form of the IL-2R will respond promptly to a 10-100-fold-lower amount of IL-2 than the cell population expressing the dimeric βγ IL-2R. Indeed, regulatory CD4^+^ T cells and activated T cells display high CD25 expression and respond to IL2 very efficiently [39,40]. On the other hand, CD8^+^ memory T cells and NK cells, which express IL-2Rβγ^+^, require higher doses of IL-2 than regulatory T cells. Upon binding to IL-2R, the signal transduction cascade is initiated through the heterodimerization of the β and γ subunits, leading to the activation of Janus kinases (JAK1 and JAK3) and the subsequent phosphorylation of the β chain. This activation recruits and phosphorylates STAT transcription factors, primarily STAT5, which dimerize, translocate to the nucleus, and bind to DNA to promote the transcription of genes responsible for T- and NK-cell activation and proliferation. This signaling pathway also engages downstream PI3K/AKT/mTOR and MAPK/ERK pathways [37,38,41,42].

High-dose IL-2 regimens have shown considerable efficacy, inducing potent cytolytic responses and the expansion of CD8^+^ T and NK cells in approximately 10% of patients. However, a substantial portion of patients experienced severe adverse effects. Among these is vascular leak syndrome (VLS) and severe pulmonary edema, attributed to direct endothelial cell action inducing vasodilation, as well as fever, hypotension, cytopenia, and organ dysfunction [43,44,45,46,47]. Both innate and adaptive immunity influence VLS onset, with NK cells contributing and Treg cells mitigating severity [48,49,50,51]. While systemic IL-2 administration for cancer has resulted in toxicity, local application, particularly intratumoral administration, shows promise [52,53,54]. However, it is evident that IL-2 can play a dual role in anti-tumor immunity. On one hand, it can trigger the generation of anti-tumor cytotoxic effector cells, such as CD8^+^ T-cell antigen-specific lymphocytes and NK cells, on the other hand, it can stimulate the immunomodulatory function of Treg cells, leading to the inhibition and exhaustion of the anti-tumor immune response. Notably, the use of low doses of IL-2 preferentially activates Treg cells, resulting in the inhibition and weakening of anti-cancer cytolytic responses [55,56,57,58,59].

Additionally, the half-life and the efficacy of IL-2 should be enhanced. It is crucial that these modified properties of IL-2 do not lead to increased vascular toxicity or heightened immunoregulatory effects [60,61]. The features of chimeric cytokines, such as IL-2 and IL-15, have been extensively described in recent studies [62]. Indeed, the generation of several mutated IL-2 as well as the pegylation or production of IL-2 prodrugs can increase the half-life and the efficacy of IL-2 as detailed [62]. The primary challenge to address is targeting IL-2 to the tumor to prolong its effect at this specific site; this to limit IL-2’s interaction with the α-chain of IL-2R on endothelial and pulmonary cells, which can lead to severe, life-threatening effects [63,64,65,66]. The solution of this issue lies in targeting the tumor with IL-2 in association with an anti-tumor-specific antibody generating an immunocytokine.

A detailed list of immunocytokines is reported by Ren et al. [67,68,69,70,71,72,73] and some of these immunocytokines will be discussed later on in this review. It is important to note that the antibody can be directed either to tumor cells or to components of the TME, such as mesenchymal stromal cells or tumor-associated fibroblasts. This targeting could enhance the local immune response by activating effector cells, such as CD8^+^ T cells and NK cells. Additionally, IL-2 can be part of a fused IT, as described in the next paragraph [62,74,75].

## 3. IL-2 Immunotoxins as Antibody–Drug Conjugates to Target and Fight Cancer Cells

The targeting of a tumor with a cytotoxic drug is one of the easiest ways to kill proliferating tumor cells [76,77,78]. This targeting is more specific when the cytotoxic drug is linked to a molecule able to recognize mainly the tumor cell [79]. This is achieved using antibody–drug conjugates (ADCs) or proteins generated by molecular engineering, which are composed of a toxin and an antibody fragment capable of interacting with a tumor-specific or tumor-associated antigen [80,81,82,83,84]. In this way, the protein carrying the toxin can enter the tumor cell, allowing the toxin to specifically kill the cell [85,86,87]. For example, it has been shown that a fusion protein, composed of the truncated diphtheria toxin, IL-2, and an anti-CCR4 antibody generating a bispecific IL2-CCR4 IT, can be used for the treatment of cutaneous T-cell lymphoma (CTCL) [88]. CTCL can indeed express good levels of IL-2Rα and CCR4 antigens, making these two receptors the optimal targets for delivering this IT. Importantly, this complex has demonstrated significant efficacy in prolonging the survival of mice bearing CTCL tumors. Notably, it was found that this IT was more effective than the anti-CD30 ADC, Brentuximab Vedotin. Interestingly, the combo of this ADC with the IT has been shown to produce a stronger anti-CTCL response than using either the ADC or the IT alone (Figure 1).

This indicates that the combination of these two drugs can enhance the therapeutic response in this murine model [89]. It is of note that the IT can target not only CTCL cells but also the infiltrating Treg cells, decreasing their immunosuppressive effects. This is an example of how an IL-2 IT can influence the anti-tumor response. Some recent data indicate that the CCR4-IL2 bispecific IT has a favorable safety profile in rats and minipig animal models, suggesting its possible future use in human clinical trials [90].

However, it is important to consider that IL-2-based IT may also interact with anti-tumor effector cells. In this context, it has been shown that Denileukin Difititox (DD), which is composed of IL-2 fused to the diphtheria toxin, can enhance anti-tumor immunity by interacting with the IL-2Rαβγ trimeric form selectively expressed on Treg cells, leading to their depletion [91,92]. These findings have been challenged by some observations indicating that DD administration in cynomolgus monkeys can lead to a significant and long-lasting depletion of peripheral blood CD16^+^CD8^+^NKG2A^+^CD3^−^ NK cells, in addition to a strong but transient elimination of peripheral Treg cells [93]. However, this depletion can be considered an unwanted side effect, as activated NK cells are important anti-tumor effectors. Interestingly, the co-administration of DD and IL-15 has been shown to prevent DD-induced NK-cell depletion both in vitro and in vivo, without affecting the strong depletion of Treg cells. This protective effect is attributed to the interaction of IL-15 with IL-2Rβγ on NK cells [91,92]. Taken together, these findings suggest that the use of IL-2-based IT should be thoroughly analyzed, with consideration given to the potential use of other factors, such as IL-15, to modulate their effects.

## 4. Immunotherapy Using Combined Tumor Microenvironment-Targeted IL-2 Cytokine

It quickly became evident that new biotechnological tools were needed for the clinical application of IL-2 to overcome important negative factors in cancer therapy. As a result, various IL-2 immunocytokines, which consist of IL-2 fused to antibodies targeting different tumor or TME antigens, have been developed and evaluated in tumor mouse models, yielding encouraging results [93,94,95,96,97,98,99,100,101,102,103]. Antibody–cytokine protein fusion combinations show promise in enhancing efficacy, when used in combination with other treatments at the appropriate time. While pro-inflammatory immunocytokines, including IL-2, can trigger side effects, these are generally of a low intensity and can be managed through careful dosing and infusion strategies. Various cytokines, such as TNF-α, IL-12, and IL-2, have been employed as they can stimulate different immune cell subsets [104,105,106,107,108,109,110]. By fusing IL-2 with antibodies targeting tumor-associated antigens, researchers aimed to localize cytokine activity to the TME, employing diverse targeting strategies and modulating pharmacokinetic properties with different antibody formats [111]. Clinical studies using IL-2 immunocytokines, particularly in combination setting with other biopharmaceuticals, have demonstrated efficacy in inducing anti-tumor immune responses mediated by CD8^+^ T cells, NK cells, and macrophages (https://www.clinicaltrials.gov accessed on 1 September 2024), as recently reviewed by Raeber et al. [112].

Diverse tumor or tumor stroma antigens have been targeted with the IL-2 immunocytokine, including the carcinoembryonic antigen (CEA), CD20, CD30, disialoganglioside 2 (GD2), epithelial cell adhesion molecule (EpCAM), fibroblast activation protein α(FAP), and extracellular matrix (ECM) proteins preferentially expressed in the tumor vasculature, such as extra domain A of fibronectin (A-FN), extra domain B of fibronectin (B-FN), and tenascin-C. The selection of ECM tumor proteins is based on the fact that some isoforms are highly expressed in the TME while they are either absent or weakly expressed in healthy tissues [113,114]. An alternative approach to using IL-2 TME specific therapy is the targeting of IL-2 to CTLs and NK cells via the NKG2D surface molecule. For this purpose, a mutant moiety of IL-2, which lacks the binding capacity to CD25, was developed. This construct is formed by using a virally high-affinity encoded NKG2D ligand, known as the orthopoxvirus major histocompatibility complex class I like protein (OMCP) [115]. It was shown that OMCP-mutIL-2 in a Lewis lung carcinoma mouse model was effective in inducing strong NK-cell-mediated therapeutic responses.

One of the most studied IL-2 immunocytokines has been L19-IL-2, targeting the neoangiogenesis tumor-specific B-FN isoform. L19-IL-2 has been evaluated in various tumor contexts in mouse models, where it has shown promising results in combination with other anti-tumor therapies. For example, when combined with anti-CTLA-4 ICIs or another immunocytokines consisting of L19 fused with TNF-α, it led to complete tumor eradication in teratocarcinoma and colon carcinoma mouse models [98]. Similarly, complete remission was achieved in the treatment of B-cell lymphoma in a xenograft mouse model with L19-IL-2 in combination with rituximab (anti-CD20 monoclonal antibody, mAb) [96]. Additionally, Cazzamalli et al. reported interesting preclinical data showing that L19-IL-2, in combination with a small-molecule drug conjugate selectively targeting cancer cells expressing carbonic anhydrase IX, exhibited high specificity and efficacy [116].

Other interesting preclinical results involving L19-IL-2 demonstrated its cooperation with various chemotherapeutics, as well as its effectiveness when intratumorally injected in melanoma and sarcoma mouse models [99]. Building on these promising findings, a phase II clinical trial was initiated for advanced melanoma patients, testing a single injection of L19-IL-2 either alone or in combination with L19-TNF [117].

Concerning IL-2 immunocytokines combined with anti-angiogenic treatments, the critical role of the cell surface heparan sulfate proteoglycan syndecan-1 (SDC1, CD138) in neovascularization, vasculogenic mimicry (VM), and tumor progression is well established. Its ectodomain could be made soluble through the action of the ADAM17 enzyme, and it acts as a coreceptor for several angiogenic molecules (VEGF, FGF-2, and others), or it could be physically associated with VEGFR-2 [118,119]. In a xenograft melanoma model, it has been demonstrated that the combination of L19-IL-2 with anti-SDC1 OC-46F2 mAb resulted in the complete inhibition of melanoma growth for up to three months after tumor injection in 71% of the treated mice [96]. More recently, in an ovarian carcinoma model, it has been confirmed that SDC1 and the tumor angiogenic B-FN isoform play pivotal roles in VM and that combined treatment using L19-IL2 and an anti-SDC1 46F2SIP antibody was effective in reducing the expression of EMT markers and loss of cancer stemness traits, which were correlated with the inhibition of VM in mice [119].

However, several research groups have made significant efforts to obtain compounds with a modified IL-2 binding capacity, i.e., specifically by designing sequences that retain binding to the IL-2Rβ and IL-2Rγ (CD122 and CD132, respectively), while eliminating binding to IL-2Rα (CD25), with the goal of minimizing toxicity. Levin and colleagues developed an engineered IL-2, referred to as a “superkine”, which has an enhanced binding affinity for IL-2βR [120], leading to a superior induction of CTL with minimal activation of Treg cells. Similarly, Sun et al. constructed an IL-2 immunocytokine comprising a tumor-targeting antibody with a super mutant IL-2 (sumIL-2), which has enhanced binding to CD122 and weak binding to CD25 [121]. The sumIL-2 is incorporated in place of one Fab of an anti-EGFR mAb, resulting in the construct Erb-sumIL2. Importantly, the combination of PD-L1 ICI with sumIL-2 therapy has shown a synergistic effect when controlling advanced tumors.

Another different strategy was developed by the Doberstein team [122]. Indeed, IL-2 with a high binding affinity for CD122 and low binding affinity for CD25 was produced, conserving the same amino acid sequence of wild-type (WT) IL-2. This IL-2 has been conjugated to multiple releasable polyethylene glycol (PEG) chains (a total of six chains), which effectively rendered this new IL-2 prodrug an IL-2Rβ agonist. This design, named Bempegaldesleukin (or NKTR-214), leverages the molecular location of the PEG molecules to achieve the selective stimulation of CD122, enhancing its therapeutic potential [122].

As of 2023, no IL-2-enhanced compounds had been approved for the treatment of cancer patients, with the exception of NKTR-214. NKTR-214 completed phase 3 studies but failed to meet its primary endpoints for both metastatic melanoma (PIVOT IO-001 trial) and advanced RCC (PIVOT-09) [111].

Klein and colleagues developed an IL-2 variant (IL-2v) immunocytokine that lacks the IL-2Rα binding capacity. This IL-2v is fused with a mAb that is specific to CEA or FAP, and it contains an inert Fc domain that does not have the ability to activate FcγR-bearing cells [101,123]. The IL2v part is derived through a structure-based mutation of key residues in the CD25 subunit of IL2R to prevent binding to IL-2Rα while preserving its affinity for IL-2Rβγ. Notably, both compounds used in combination therapy demonstrated superior activity compared to conventional IL-2-based immunocytokines. These engineered compounds were able to trigger and reinforce NK-cell and CD8^+^ effector T-cell activation via IL-2Rβγ, both in the periphery and within the TME. Furthermore, they synergized with anti-PD-L1 ICI therapy, as well as with trastuzumab and cetuximab chemotherapies, enhancing the overall anti-tumor response [101,123].

The in vivo anti-cancer immune response of FAP-IL2v in various combination immunotherapies was investigated in both xenograft and syngeneic murine tumor models. FAP has a low expression in or is absent from healthy adult tissues, but is highly expressed in cancer-associated fibroblasts and pericytes in the majority of human epithelial cancers.

When combined with different therapeutic antibodies able to trigger antibody-dependent cellular cytotoxicity (ADCC), such as the anti-EGFR antibody cetuximab, FAP-IL2v significantly extended the median survival compared to single-agent treatments of BALB-neuT genetically engineered mice that spontaneously developed breast tumors. Notably, mice that responded to this therapy showed increased tumor infiltration by CD3^+^ T cells, NK cells, and CD68^+^ macrophages, highlighting the enhanced immune response facilitated by the combination therapy [123].

## 5. Therapeutic Relevance of Enhancing Effector Cytolytic CD8^+^ T-Cell Responses Induced by the Combination of Immunocytokine IL-2 with PD-1 cis-Targeting

A key objective of cancer immunotherapy is the activation of non-exhausted, functional cytolytic CD8^+^ T effector responses, alongside the activation of NK effector cells. T-cell exhaustion is often associated with the expression of markers, such as PD-1, TIM-3, and CD39, which limit effective anti-tumor immunity [124,125,126,127]. A promising therapeutic strategy to overcome T-cell exhaustion was demonstrated with the experiments reported by Klein’s group [128]. Indeed, an IL-2v immunocytokine targeting PD-1 in cis (named PD1-IL-2v) and directed to the FAP tumor stroma antigen has been developed [128]. The IL-2v variant avoids binding to CD25, thus limiting Treg cell expansion and reducing the immunosuppressive effects commonly observed with WT IL-2 therapy. Importantly, this modification enhanced the functional activation of the stem-like CD8^+^ T cells (TCF1^+^ PD-1^+^), a subset critical for durable immune responses. This effect is mediated by strong STAT5 phosphorylation, which boosts the activation of these T cells.

Similarly, murine PD-1-IL-2v (muPD-1-IL-2v) also induced more potent CD8^+^ T-cell effectors in the C57/BL6 model in a syngeneic pancreatic ductal adenocarcinoma model, Panc02 [128]. This CD8^+^ T-cell effector population can induce GM-CSF production, as well as a higher concentration of granzyme B. It is notable that GM-CSF is a cytokine involved in dendritic cell activation and in potentiating T-cell cytotoxic effector functions [129]. Interestingly, the highest therapeutic efficacy of muPD-1-IL-2v was further demonstrated in several in vivo mouse models, such as MCA205, B16-F10-OVA, and RipTag5 pancreatic neuroendocrine tumors refractory to ICIs [130]. Furthermore, Tichet et al. showed that muPD1-IL-2v was able to induce remarkable tumor regression and increased survival in combination with anti-PD-L1 [128]. Notably, this therapeutic combo generated and expanded polyfunctional effector memory CD8^+^ TILs with significantly higher levels of granzyme B, IFNγ, and TNFα compared to CD8^+^ TILs isolated from mice treated with only muPD1-IL2v. This IL-2v immunocytokine entered clinical trials in 2020 as both a monotherapy and in combination with anti-PD-L1 antibodies. This approach appears to be the most promising for all those patients who have developed an endogenous T-cell response to the tumor. Also, it is widely accepted that, to obtain high and superior therapeutic responses, in addition to stem-like CD8^+^ T-cell-mediated responses, innate NK-type effectors and M1-type macrophages are also necessary. Indeed, the combination treatment with anti-PD-L1 and muPD1-IL2v supported the immune response by reprogramming tumor-associated macrophages and increasing the diversity of the immune repertoire [128]. In particular, single-cell RNA sequence analyses of myeloid cells showed that, in tumor-bearing mice treated with the aforementioned combination therapy, there was a reduction in the expression of M2-like immunosuppressive markers, such as Trem2, Mrc1, and CD163, alongside an upregulation of pro-inflammatory markers, such as Irf1, Slamf8, IFNγ, as well as the presence of T-cell chemokines, like as CXCL9 and CXCL10 [128]. Furthermore, in the orthotopic glioma model with the GL261 cells, it has been reported that combination treatment can increase stem-like PD-1^+^TCF-1^+^CD8^+^ T cells, as well as their progeny, PD-1^+^TCF-1^−^, and effector CD8^+^ T cells. In this model, CD8^+^ T cells isolated from mice treated with the combination were less suppressive than those isolated from mice treated with either of the two drugs alone. Again, a shift from M2 to M1 macrophages was detected. These findings suggest that, in several different mouse models, the combination of ICIs and IL-2 variants can contribute to an optimal anti-tumor immune response [128]. Besides the effects described, it is worth noting that combination treatment can affect the phenotype of endothelial cells present within the tumor. Indeed, these endothelial cells can directly regulate the infiltration, proliferation, and cytotoxicity of CD8^+^ T cells in an antigen-specific manner [131,132].

## 6. Role of Radiotherapy in Triggering Anti-Tumor Therapeutic Responses in Combined Tumor-Targeted IL-2 Treatments

Recent research has shown a growing interest in investigating the synergistic effects of combining radiotherapy (RT) with tumor-targeted IL-2 treatments to eradicate tumors. Indeed, the relationship between RT and the modulation of the immune system’s response have revealed novel regulatory pathways [133,134]. These include the concept of radiation-induced tumor equilibrium (RITE), which serves as a starting point to discuss the mechanistic influence of immune checkpoint therapies on the efficacy of RT [133,134].

Traditionally viewed as a local therapy, RT directly damages DNA in tumor cells [135]. RT usually induces the release of danger signals and consequently chemokines that recruit inflammatory cells into the TME, including antigen-presenting cells that activate the cytotoxic T-cell function. However, RT also has the capability to attract immunosuppressive cells into the TME [136]. Consequently, it has been shown that RT exhibits both immunostimulatory and immunosuppressive properties, owing to the inherent sensitivity of immune cells to its effects [137,138].

Concerning the immunostimulatory effects, localized RT can provoke systemic immune responses [139] by inducing the expression of tumor-associated antigens and generating new tumor antigens that can activate anti-tumor immune responses. This helps counter the tumor’s suppression of antigen presentation. For example, the expression of MHC-I, a critical antigen recognized by CD8^+^ antigen-specific T cells, is often decreased on tumor cells [140]. RT can effectively increase MHC-I expression, facilitate dendritic cell maturation and leukocyte infiltration of tumors [141], reduce the presence of Treg cells within tumors, expand T-cell populations, and enhance T-cell migration. Moreover, RT has demonstrated the capability to convert non-immunogenic tumors into immunogenic ones, either partially or completely [142]. Previous studies using the IL-2 immunocytokine, such as L19-IL-2, have shown that it can synergize with RT to achieve tumor rejection and an immune-mediated protection effect in a colon carcinoma mouse model [143,144,145]. Additionally, RT can manage local tumor progression when combined with anti-PD1/PDL1 or anti-CTLA4 ICI, immunocytokines, dendritic cell vaccines, and Toll-like receptor antagonists. These therapeutic combinations improve the overall survival (OS) and elicits specific immune responses against cancer [146]. Some recent preclinical and clinical research have demonstrated that the combination of RT’s immunosuppressive and immunostimulatory properties can lead to abscopal effects [147] and potentially a “radio-memory” effect, where the synergistic action of combo treatments produces enhanced outcomes [148].

Investigating whether the “radio-memory” effect [149] extends beyond anti-PD1/PDL1 ICI to other forms of immune therapy, such as IL-2, is a critical step in understanding the broader implications of RT-induced immune responses in cancer treatment. For instance, in patients with malignant pleural effusions who had previously received RT for non-small-cell lung cancer (NSCLC) within 18 months, the intrapleural infusion of IL-2 or cisplatin demonstrated potential evidence of a radio-memory effect [149].

More recently, other studies have examined the critical role of IL-2 in synergistic immunotherapeutic strategies involving RT. In particular, a modified IL-2-based treatment has shown promising results when combined with RT to amplify immune responses. Gadwa and colleagues demonstrated that combining RT with PD1-IL2v, an IL-2 variant targeting intermediate-affinity IL-2 receptors (IL-2Rβγ), significantly enhanced immune activation. The lymphocytes involved are mainly CD8^+^ T cells and NK cells, while a concomitant reduction in the suppressive function of Treg cells was detected [150]. This combination showed effective tumor control and reduced metastasis, leveraging the immune system’s response to control distant tumors via an “abscopal effect”. In a parallel investigation, He et al. focused on an engineered IL-2 variant, RDB 1462, which binds selectively to IL-2Rβγ, promoting anti-tumor activity while minimizing the induction of Tregs [151]. Their findings revealed that combining high-dose RT with RDB 1462 significantly improved survival and suppressed primary and metastatic tumor growth in mouse models. Moreover, this combination therapy enhanced CD8^+^ memory T cells and NK-cell activation, while diminishing Treg and MDSCs. The precise timing of IL-2 administration before RT was crucial for maximizing the therapeutic efficacy and immune response [151]. These findings demonstrate that IL-2, particularly in its engineered forms, holds substantial promise when paired with RT to enhance anti-tumor immunity. By targeting specific IL-2 receptor subunits, these treatments can potentially overcome immune suppression within the TME and improve the therapeutic outcomes of RT and other immunotherapies. Further clinical investigations are warranted to optimize this approach for broader cancer treatment strategies (Figure 2).

## 7. Comparative Analysis of IL-2, IL-15, and IL-21 in Cancer Immunotherapy

IL-2 is a key cytokine in cancer immunotherapy, especially its ability to stimulate the expansion of NK cells and T lymphocytes. This property makes it integral to adoptive transfer protocols designed to enhance lymphocyte culture and persistence in cancer patients. High-dose IL-2 infusion is FDA-approved for metastatic RCC and melanoma treatment. However, the systemic administration of IL-2 at the recommended dose often leads to severe toxicities, including grade 3 and 4 adverse effects, which limit its clinical use. As a result, second-generation IL-2 therapies have been developed to address these issues by enhancing both pharmacokinetics and pharmacodynamics [152] (Table 1). This table compares different properties of IL-2, IL-15, and IL-21, and their application is described in detail further.

To improve IL-2’s pharmacokinetic profile, strategies have focused on extending its circulation half-life. This is achieved by attaching the cytokine to molecules like the Fc domain of immunoglobulins or polyethylene glycol (PEG), or by chimerizing IL-2 with antibodies targeting the TME. These modifications help improve its stability and efficacy. In terms of pharmacodynamics, modifications reduce IL-2’s binding to the high-affinity IL-2 receptor (IL-2Rα), which is predominantly expressed on regulatory T (Treg) cells. This adjustment allows IL-2 to preferentially stimulate NK and T cells, which are critical for anti-tumor immunity, while limiting its ability to expand Tregs that can suppress immune responses [101,153] (Table 1).

Engineered IL-2 variants, such as NKTR-214, modify IL-2 with PEG molecules to create a longer-acting, inactive form that eventually activates and interacts with the medium-affinity IL-2 receptor. Clinical trials combining NKTR-214 with immune checkpoint inhibitors, including nivolumab and atezolizumab, have shown promising results in melanoma, RCC, and NSCLC. While the combination therapy has been well-tolerated, further randomized studies are planned to confirm its benefits over monotherapy [153,154].

Another approach to improving IL-2’s therapeutic profile involves engineering mutated variants with a reduced affinity for IL-2Rα, such as cergutuzumab amunaleukin, which fuses a mutated IL-2 with an antibody targeting carcinoembryonic antigen (CEA). These fusion proteins are being tested in clinical trials with various therapeutic agents, demonstrating potential in combination treatments with drugs like trastuzumab and atezolizumab [101].

IL-15, another important cytokine, has shown considerable promise in cancer immunotherapy due to its role in supporting NK- and CD8^+^ T-cell proliferation without stimulating Tregs. Unlike IL-2, IL-15 does not bind to IL-2Rα, thus avoiding Treg activation and its associated immunosuppressive effects. Initial clinical trials with recombinant IL-15 demonstrated the expansion of NK and CD8^+^ T cells in patients with advanced melanoma and RCC. However, severe adverse effects, such as fever and hypotension, led to the halting of some trials at lower doses [155,156,157,158] (Table 1).

One major issue with IL-15 is its reliance on IL-15Rα for signaling, which compromises its stability. To address this, several engineered IL-15 variants have been developed, such as hetIL-15 and hetIL-15Fc. These modifications aim to improve its half-life and stability, enabling prolonged NK- and T-cell expansion. hetIL-15, based on the natural heterodimeric state of IL-15 and IL-15Rα, has shown promising preclinical results and is undergoing clinical trials for treating metastatic and unresectable solid tumors. hetIL-15Fc, a glycosylated version of IL-15 fused with the Fc region of human IgG1, also demonstrates enhanced efficacy in murine models, supporting the benefits of improved stability for sustained immune activation [159,160,161,162,163].

N-803, another engineered IL-15 superagonist, combines IL-15 with an IL-15Rα sushi domain and an Fc fragment, resulting in a remarkable half-life and bioactivity. Preclinical studies suggest that N-803 enhances NK-cell cytotoxicity and can eliminate established tumors. Clinical trials have confirmed its tolerability and efficacy, positioning N-803 as a promising candidate for advanced cancer treatments [164,165,166].

ALT-803, another engineered version of IL-15, fuses IL-15 with the IL-15Rα sushi domain and an IgG1 Fc domain, improving its half-life and anti-tumor effects. Clinical trials of ALT-803 have shown increased NK- and CD8^+^ T-cell expansion in patients with hematological cancers and metastatic NSCLC, with good tolerability, especially in combination with PD-1 inhibitors. Its promising results and fast-track designation from the FDA for bladder cancer treatment highlight IL-15’s potential [167,168,169,170].

Engineered IL-15 therapies offer several advantages over IL-2, primarily in reducing toxicity while enhancing immune cell expansion. Unlike IL-2, which stimulates both effector immune cells and Tregs, IL-15 preferentially expands NK and memory T cells, leading to more selective anti-tumor activity. IL-15 has also been successfully incorporated into adoptive cell therapies, including CAR-T cells, where it improves cell expansion, persistence, and efficacy. Additionally, IL-15 is being tested in combination with oncolytic viruses and tumor-conditional IL-15 pro-cytokines to induce localized immune expansion with minimal toxicity. Despite these advancements, safety concerns remain, requiring careful evaluation in clinical trials [155,171,172] (Table 1).

IL-21, a member of the IL-2 family, is another cytokine under investigation in cancer immunotherapy. Like IL-2, IL-21 shares a common gamma receptor (γc) with IL-2 and IL-15, influencing the activation and proliferation of immune cells, including T cells and B cells. IL-21 has garnered attention for its ability to stimulate B-cell differentiation into plasma cells, enhance immunoglobulin production, regulate CD4^+^ and CD8^+^ T-cell responses, and limit Treg differentiation. This makes it a promising candidate for enhancing anti-tumor immunity, similar to IL-2’s effects on T and NK cells [173] (Table 1).

However, despite these similarities, the clinical development of IL-21 as an anti-cancer therapy has progressed more slowly than IL-2. IL-2, where its proven success in treatments for melanoma and RCC has seen extensive use in cancer immunotherapy and significant clinical trials. In contrast, IL-21’s clinical trials are still in the early stages. While IL-21 has been tested alone and in combination with therapies such as ipilimumab, nivolumab, sunitinib, rituximab, sorafenib, and doxorubicin (e.g., NCT00095108 and NCT01489059), its progress remains modest. In a phase I dose-escalation trial, only 3 out of 26 patients experienced partial responses, although pharmacodynamic effects on tumor immunity were observed. Additionally, when combined with rituximab in patients with non-Hodgkin lymphoma, IL-21 showed clinical responses in 8 out of 19 patients [174,175,176] (Table 1).

Unlike IL-2, which is already in use for cancer treatment, IL-21’s clinical testing is still in its infancy, with only 14 clinical trials conducted since its development began in 2004. These trials have primarily focused on hematological cancers, melanoma, and RCC. In some cases, IL-21 is being tested as a fusion protein with albumin to improve its pharmacokinetics, though the natural form of IL-21 is more commonly used [177] (Table 1). While IL-2 has established itself as a cornerstone of cancer immunotherapy, the potential of IL-21 is still being explored, particularly in combination with other treatments. Given its effects on both T cells and B cells, as well as its ability to limit Treg differentiation, IL-21 holds promise as a future therapeutic option, especially in combination regimens. However, its development lags behind IL-2, and further research is needed to determine its full clinical potential.

## 8. In Situ Tumor Vaccination with IL-2 or IL-2 Immunocytokines

To overcome the immunomodulatory properties of the TME, the use of in situ tumoral vaccinations has been proposed [178]. This therapeutic strategy targets the relieving of immunotolerance elicited locally by the growing tumor using appropriate drugs as adjuvants, such as Toll receptor ligands and/or immunocytokines [178]. The intratumoral immunization and the induction of a strong immune response in a poorly immunogenic environment have been well-reviewed recently [178,179,180,181]. Herein, we report some relevant models of the therapeutic approaches supporting the evidence that this kind of vaccination can elicit the desired anti-tumor response.

In preclinical murine models of melanoma, head and neck carcinoma, and neuroblastoma, it has been shown that the intratumoral injection of an immunocytokine composed of Hu14.18-IL2 can result in the complete regression of well-established tumors together with a tumor-specific T-cell memory [182]. In this instance, the immunocytokine consists of human IL2 genetically fused to each IgG heavy chain of the GD2 mAb Hu14.18. Hu14.18-IL2 induced a strong cooperative effect with radiation [182], and this effect was not reduced by the depletion of NK cells but by T cells. Furthermore, the combo of radiation, hu14.18-IL.2, and anti-CTLA-4 improved animal survival and reduced the burden of tumor metastases compared to the use of radiation and immunocytokines [182]. This finding shows that a T-cell-specific immune response has been elicited that can be further enhanced by an immune checkpoint blockade.

Utilizing bilateral flank tumor model, it has been reported that the intratumoral injection of the membrane-tethered IL-2-armed oncolytic virus (vvDD-mIL2) plus CpG oligodeoxynucleotide can trigger systemic immunization. This immunization yielded the reduction of a contralateral untreated tumor [183]. This anti-tumoral activity was dependent on CD8^+^ T cells and IFNγ production, but not CD4^+^ T cells. A further improvement of the anti-tumoral response in the non-injected tumor was detected by the inhibition/elimination of suppressive components, such as tumor-associated macrophages with clodronate [183].

However, a key role in generating an efficient anti-tumor response for NK cells has been shown recently [184]. Indeed, radiotherapy followed by the intratumoral injection of IL-2 and a tumor-specific monoclonal antibody into the tumor irradiated site (3xTx) can increase the in situ vaccination by recruiting NK cells. The depletion of NK cells reduced the polyfunctionality of CD8^+^ T cells. Also, the induction of CD86 expression on NK cells associated with the natural killer group 2 (NKG2) D activating receptor function was involved in the apoptosis of CTLA4^+^ Tregs in the TME [184]. This effect triggers the propagation of anti-tumor-specific immunity throughout the mouse body following 3xTx “in situ” vaccination [184]. Importantly, in this murine model, IL-2 was administered locally to avoid its systemic side effects, while the anti-tumor-specific anti-GD2 mAb Hu14.18 could be administered systemically [184]. Finally, the intratumoral injection with IL-2 and a multifunctional nanoparticle (PIC) has been reported to improve the anti-tumor response [185]. In detail, the PIC can be considered as an adjuvant composed of a scalable and simple complexation of poly-L-lysine, CpG oligodeoxynucleotide, and an iron oxide nanoparticle. The triple-combo of RT + IL-2 + PIC significantly improved the CD8 T-cell-mediated immune response surpassing the single or dual combination of these treatments [185]. Altogether, these findings strongly support IL-2 or IL-2 derivatives in combo therapies to augment the in situ vaccine effect of RT in clinical settings.

## 9. Conclusions

The integration of IL-2 immunocytokines into cancer therapy represents a significant advancement, addressing many of the challenges associated with traditional high-dose IL-2 treatment. While high-dose IL-2 has demonstrated durable efficacy in a subset of patients with metastatic melanoma and RCC, its severe toxicities, such as vascular leak syndrome and cytokine release syndrome, have limited its broader clinical application. The emergence of targeted IL-2 delivery strategies, including immunotoxins, immunocytokines, and engineered variants, has helped mitigate these issues by improving specificity to the tumor microenvironment, reducing off-target effects, and enhancing therapeutic outcomes.

Innovative approaches, such as antibody fusion proteins, bi-specific formats, and modified IL-2 variants, have significantly optimized the delivery and efficacy of IL-2 therapies. For example, PD-1-targeting IL-2 immunocytokines have shown the ability to rejuvenate exhausted T cells, improve immune infiltration into tumors, and synergize with checkpoint inhibitors to counteract immune suppression. These advances, combined with cytokine engineering techniques, like glyco-engineering and prodrug formulations, have enhanced the selectivity and cost-effectiveness of IL-2-based therapies, enabling their broader use across diverse clinical contexts. Furthermore, the combination of IL-2 immunocytokines with checkpoint inhibitors, chemotherapy, radiotherapy, and anti-angiogenic agents has elicited robust anti-tumor responses in both preclinical and early clinical trials, effectively improving tumor clearance while minimizing toxicity.

Despite these advancements, several IL-2-based therapies have failed to meet the clinical endpoints, providing critical insights for future development. High-dose IL-2 therapies, such as Proleukin^®^, have demonstrated remarkable responses in highly selected patient subsets, but often lack efficacy in broader populations due to variability in tumor biology and immune landscapes. Toxicity-related discontinuations remain a significant challenge, with severe adverse effects limiting the doses that can be administered. For example, bempegaldesleukin (NKTR-214), an IL-2 variant engineered to preferentially activate effector T cells, failed to meet the endpoints in some trials due to insufficient efficacy as a monotherapy or limited synergy in combination regimens. Additionally, certain IL-2 therapies have inadvertently expanded regulatory T cells, which suppress anti-tumor immunity, particularly when lacking strategies to selectively target effector T cells. In some cases, therapies have struggled to effectively overcome immune cold tumor microenvironments, limiting efficacy in metastatic or poorly infiltrated tumors.

Looking forward, further advances in IL-2 immunocytokines offer immense potential to address these challenges. Enhancing tumor-specific targeting through innovations in antibody–cytokine fusion proteins and multi-specific formats can increase therapeutic concentrations within tumors while minimizing systemic exposure. Optimizing combination regimens with immune checkpoint inhibitors and other therapies will help harness a synergy to overcome resistance and immune suppression. Reducing toxicities through conditional activation and prodrug designs will broaden the patient populations eligible for IL-2-based therapies. Addressing heterogeneous tumor responses through stratified clinical trial designs, which account for differences in immune status, tumor immunogenicity, and genetic markers, will identify those most likely to benefit from these therapies.

The ability to modulate immune responses precisely, coupled with the integration of IL-2 immunocytokines into tailored treatment strategies, has the potential to redefine cancer therapy. Continued research into the mechanisms of action, combination strategies, and patient selection criteria will be pivotal to unlocking the full clinical potential of these therapies. With sustained innovation and rigorous clinical evaluation, IL-2 immunocytokines could provide durable responses and significantly improve survival outcomes for patients across a wide range of cancers.

## Figures and Tables

**Figure 1 vaccines-13-00069-f001:**
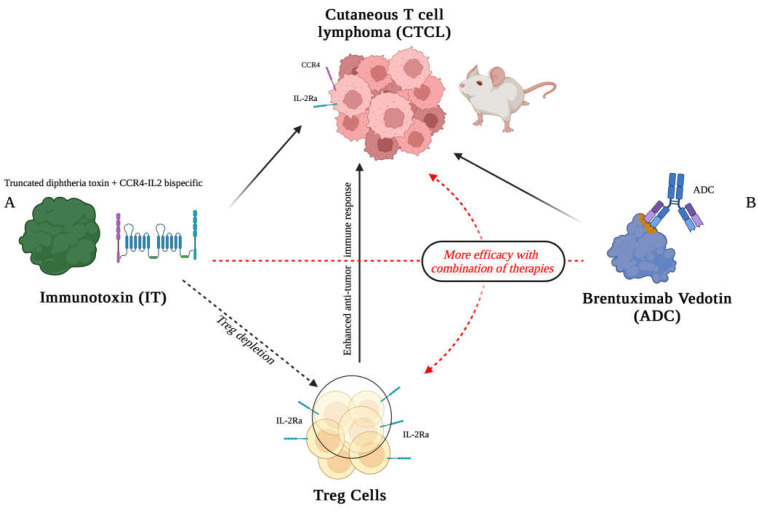
Schematic representation of the therapeutic action of a bispecific immunotoxin (IT) targeting CCR4 and IL-2Rα and the anti-CD30 antibody–drug conjugate (ADC) Brentuximab Vedotin in a cutaneous T-cell lymphoma (CTCL). The CTCL cells expressing CCR4 (purple) and IL-2Rα (green) are the key receptors targeted by the IT (**A**). The IT is composed of a truncated diphtheria toxin and an anti-CCR4 antibody and IL2. This bispecific component binds to CCR4 and IL-2Rα delivering its cytotoxic payload (the truncated diphtheria toxin). In addition to targeting CTCL cells, the IT also depletes regulatory T cells (Tregs), which express IL-2Rα, enhancing the anti-tumor immune response (dashed black arrow). On the right, Brentuximab Vedotin (**B**), an ADC targeting the CD30 antigen expressed on CTCL, can deliver the toxic drug Vedotin to tumor cells. The combination of IT and ADC therapies (red dashed arrows) leads to a synergistic effect, resulting in improved tumor suppression and survival in preclinical mouse models. Figure created with Biorender.com (accessed on 1 September 2024).

**Figure 2 vaccines-13-00069-f002:**
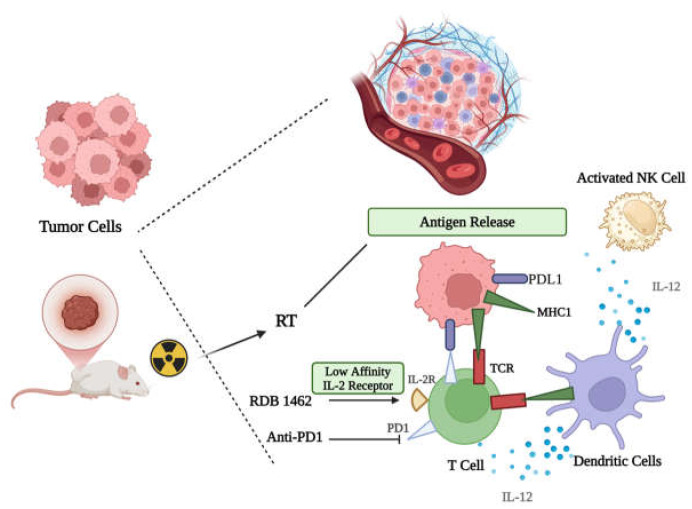
Graphical representation of the effect of radiotherapy (RT) on cancer cells and its impact on the immune response. Radiation induces the killing of tumor cells, favoring the release of tumor-associated antigens (TAAs) or neoantigens. These antigens can be presented by antigen-presenting cells (APCs), such as dendritic cells, in the context of MHC-I molecules. The complex of MHC-I and the tumor antigen epitope expressed on APCs activates the immune response through recognition by the T-cell receptor (TCR). This interaction triggers T-cell activation and initiates an anti-tumor immune response. Activated cytolytic T cells recognize and destroy tumor cells. T-cell activity can be downregulated by the interaction between PDL1 (Programmed Death-Ligand 1) expressed on tumor cells and PD1 (Programmed Death-1) on effector cells (CTLs). Therapeutic agents, such as RDB 1462 targeting the IL-2 receptor, and anti-PD1 antibodies may enhance the anti-tumor immune response. RDB 1462 further activates effector cells expressing low-affinity IL-2 receptors, while anti-PD1 antibodies block inhibitory PDL1-PD1 signaling, restoring CTL activity against tumor cells (**upper** scheme). The full activation of T cells is mediated by the interaction of CD28 on T cells and CD80/CD86 co-stimulatory molecules (**lower** scheme). The response to tumor-associated antigens can be increased by the use of agonists of Toll receptors and the in situ administration of IL-2 derivatives (see Section 8 for further explanation). The T-APC interaction enhances the release of IL-12 by APCs, a cytokine that promotes T-cell and natural killer (NK)-cell activation (upper scheme). IL-12 plays a pivotal role in skewing the immune response toward a Th1 phenotype, thereby enhancing cytotoxic activity against tumor cells. Experimental mouse models have been used to study and validate these molecular mechanisms underlying RT-induced immune activation and the efficacy of combination immunotherapies for in situ vaccination. Figure created with Biorender.com (accessed on 1 September 2024).

**Table 1 vaccines-13-00069-t001:** Comparison of cytokines in cancer immunotherapy, highlighting the mechanisms, benefits, and limitations.

Cytokine	Mechanism of Action	Advantages	Disadvantages
IL-2	Activates T cells and NK cells [37,38,41,42]	High efficacy; strong synergy with checkpoint inhibitors; engineered forms reduce toxicity [101,153,154]	Severe toxicities (e.g., VLS); short half-life [43,44,45,46,47,48,49,50,51]; activation of Treg [55,56,57,58,59]
IL-15	Promotes survival and proliferation of NK and memory T cells [91,92]	Reduced toxicity compared to IL-2; supports long-term immune memory [155,156,157,158]	Limited clinical experience; challenges with large-scale production [155,156,157,158,159,160,161,162,163,164,165,166,167,168,169,170,171,172]
IL-21	Stimulates B-cell differentiation, regulates T-cell function, and limits Treg differentiation [173,174,175,176,177]	Promotes anti-tumor immunity; potential synergy with checkpoint inhibitors; enhances adoptive T-cell therapy [173,174,175,176,177]	Clinical development in early stages; limited trials and indications; slower progression compared to IL-2 [173,174,175,176,177]

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
