# Peer review of "Efficacy of Anti-Cancer Immune Responses Elicited Using Tumor-Targeted IL-2 Cytokine and Its Derivatives in Combined Preclinical Therapies"

_vaccines, 2025, doi:10.3390/vaccines13010069_

Round 1

Reviewer 1 Report

Comments and Suggestions for Authors

The review of Mortara and coworkers explores recent advances in IL-2-based immunotherapeutic approaches for cancer treatment, emphasizing tumor-targeted strategies that address the TME while minimizing systemic toxicity. 

Next some comments that could be addressed in order to strengthen the manuscript's clarity, comprehensiveness, and critical evaluation. 

1. Please, if possible, expand the discussion on clinical trial outcomes, particularly explaining why certain IL-2-based therapies failed to meet endpoints. 

2. Please, improve figure 2 descriptions and ensure all graphics (cell and molecules) are explicitly referenced.

3. It will be nice of the authors broaden the scope of the analysis including comparisons with alternative cytokine-based therapies, highlighting advantages and disadvantages of IL-2 based ones. A table could be included in this part.

Author Response

  1. Please, if possible, expand the discussion on clinical trial outcomes, particularly explaining why certain IL-2-based therapies failed to meet endpoints. 

We appreciate the reviewer's thoughtful insight into the clinical outcomes of IL-2-based therapies. As outlined in the discussion, while IL-2 immunocytokines have demonstrated significant promise, certain therapies have failed to meet clinical endpoints due to several factors. For instance, bempegaldesleukin (NKTR-214), an IL-2 variant engineered to preferentially activate effector T cells, did not achieve expected efficacy in some clinical trials. This failure can be attributed to multiple factors, including insufficient monotherapy efficacy, limited synergy in combination regimens, and unintended expansion of regulatory T cells that can dampen the antitumor immune response. Additionally, therapies have faced challenges in overcoming the immune-cold tumor microenvironment, which can limit efficacy, particularly in metastatic or poorly infiltrated tumors.

These outcomes underscore the complexity of IL-2-based therapies, highlighting the need for careful patient selection, combination strategies, and tumor-specific targeting to optimize clinical outcomes. We believe that continued innovation and more precise clinical trial designs, which account for tumor biology and immune status, will be crucial in improving the success rate of these therapies moving forward.

Thank you for raising this important point, and we hope this clarification helps address the underlying reasons behind the clinical trial challenges observed in some IL-2-based therapies. We have modified the Discussion accordingly.

Conclusions

The integration of IL-2 immunocytokines into cancer therapy represents a significant advancement, addressing many of the challenges associated with traditional high-dose IL-2 treatment. While high-dose IL-2 has demonstrated durable efficacy in a subset of patients with metastatic melanoma and renal cell carcinoma, its severe toxicities, such as vascular leak syndrome and cytokine release syndrome, have limited its broader clinical application. The emergence of targeted IL-2 delivery strategies, including immunotoxins, immunocytokines, and engineered variants, has helped mitigate these issues by improving specificity to the tumor microenvironment, reducing off-target effects, and enhancing therapeutic outcomes. 

Innovative approaches, such as antibody fusion proteins, bi-specific formats, and modified IL-2 variants, have significantly optimized the delivery and efficacy of IL-2 therapies. For example, PD-1-targeting IL-2 immunocytokines have shown the ability to rejuvenate exhausted T cells, improve immune infiltration into tumors, and synergize with checkpoint inhibitors to counteract immune suppression. These advances, combined with cytokine engineering techniques like glyco-engineering and prodrug formulations, have enhanced the selectivity and cost-effectiveness of IL-2-based therapies, enabling their broader use across diverse clinical contexts. Furthermore, the combination of IL-2 immunocytokines with checkpoint inhibitors, chemotherapy, radiotherapy, and anti-angiogenic agents has elicited robust antitumor responses in both preclinical and early clinical trials, effectively improving tumor clearance while minimizing toxicity. 

Despite these advancements, several IL-2-based therapies have failed to meet clinical endpoints, providing critical insights for future development. High-dose IL-2 therapies, such as Proleukin®, have demonstrated remarkable responses in highly selected patient subsets but often lack efficacy in broader populations due to variability in tumor biology and immune landscapes. Toxicity-related discontinuations remain a significant challenge, with severe adverse effects limiting the doses that can be administered. For example, bempegaldesleukin (NKTR-214), an IL-2 variant engineered to preferentially activate effector T cells, failed to meet endpoints in some trials due to insufficient efficacy as a monotherapy or limited synergy in combination regimens. Additionally, certain IL-2 therapies have inadvertently expanded regulatory T cells, which suppress antitumor immunity, particularly when lacking strategies to selectively target effector T cells. In some cases, therapies have struggled to effectively overcome immune-cold tumor microenvironments, limiting efficacy in metastatic or poorly infiltrated tumors. 

Looking forward, further advances in IL-2 immunocytokines offer immense potential to address these challenges. Enhancing tumor-specific targeting through innovations in antibody-cytokine fusion proteins and multi-specific formats can increase therapeutic concentrations within tumors while minimizing systemic exposure. Optimizing combination regimens with immune checkpoint inhibitors and other therapies will help harness synergy to overcome resistance and immune suppression. Reducing toxicities through conditional activation and prodrug designs will broaden the patient populations eligible for IL-2-based therapies. Addressing heterogeneous tumor responses through stratified clinical trial designs, which account for differences in immune status, tumor immunogenicity, and genetic markers, will identify those most likely to benefit from these therapies. 

The ability to modulate immune responses precisely, coupled with the integration of IL-2 immunocytokines into tailored treatment strategies, has the potential to redefine cancer therapy. Continued research into the mechanisms of action, combination strategies, and patient selection criteria will be pivotal to unlocking the full clinical potential of these therapies. With sustained innovation and rigorous clinical evaluation, IL-2 immunocytokines could provide durable responses and significantly improve survival outcomes for patients across a wide range of cancers.

  1. Please, improve figure 2 descriptions and ensure all graphics (cell and molecules) are explicitly referenced.

Figure 2. Graphical representation of the effect of radiotherapy (RT) on cancer cells and its impact on the immune response. Radiation induces the killing of tumor cells favoring the release of tumor-associated antigens (TAAs) or neoantigens.  These antigens can be presented by antigen-presenting cells (APCs), such as dendritic cells, in the context of the MHC-I molecules. The complex of MHC-I and tumor antigen epitope expressed on antigen-presenting cells (small brown cell on the lower right) can activate the immune response through the recognition by the T cell receptor (TCR). This interaction triggers T cell activation and initiates an anti-tumor immune response. Activated cytolytic T cells (dark blue cell below) can recognize the antigen on tumor cells (red cell). The effector function can be downregulated by the interaction between the PDL1 (Programmed Death-Ligand 1) expressed on tumor cells (red cell) and PD1 (Programmed Death-1) on effector cells (CTLs). In this context, therapeutic agents such as RDB 1462 targeting IL-2 receptor and anti-PD1 antibodies may enhance the anti-tumor immune response. Indeed, RDB 1462 activates further the effector cells expressing low affinity receptors for IL-2. On the other hand, the anti-PD1 antibody blocks the negative signal mediated by the PDL1-PD1 interaction, restoring CTL activity against tumor cells. Experimental mouse models have been used to study and demonstrate these molecular mechanisms underlying RT-induced immune activation and the efficacy of combination immunotherapies. Figure created with Biorender.com (accessed on 1 September 2024).

  1. 3. It will be nice of the authors broaden the scope of the analysis including comparisons with alternative cytokine-based therapies, highlighting advantages and disadvantages of IL-2 based ones. A table could be included in this part.

Thank you for the insightful suggestion to expand the analysis by comparing IL-2-based therapies with other cytokine-based approaches. In response, we have included a comparison with IL-15 and IL-21, two promising cytokines in cancer immunotherapy. IL-15 offers advantages over IL-2 by preferentially expanding NK and memory T cells while avoiding the expansion of regulatory T cells (Tregs), which can dampen anti-tumor immunity. Engineered IL-15 variants, such as N-803 and ALT-803, have shown enhanced stability and bioactivity, making them strong candidates for cancer therapy. However, challenges related to their clinical development and large-scale production remain, limiting their use compared to IL-2.

IL-21, while sharing the common gamma receptor with IL-2 and IL-15, has a broader impact on immune cell function, including B cell differentiation and the regulation of Treg differentiation. Despite its potential, IL-21’s clinical development has progressed more slowly, and it is still in the early stages of testing compared to IL-2. We have included a table (Table 1) summarizing the mechanisms, advantages, and disadvantages of IL-2, IL-12, IL-15, and IL-21, offering a clear comparison of these cytokines in the context of cancer immunotherapy. This expanded discussion provides a more comprehensive overview of the landscape of cytokine-based therapies.

7. Comparative analysis of IL-2, IL-15, and IL-21 in cancer immunotherapy

IL-2 is a key cytokine in cancer immunotherapy, especially for its ability to stimulate the expansion of NK cells and T lymphocytes. This property makes it integral to adoptive transfer protocols designed to enhance lymphocyte culture and persistence in cancer patients. High-dose IL-2 infusion is FDA-approved for metastatic renal cell carcinoma (RCC) and melanoma treatment. However, systemic administration of IL-2 at the recommended dose often leads to severe toxicities, including grade 3 and 4 adverse effects, which limit its clinical use. As a result, second-generation IL-2 therapies have being developed to address these issues by enhancing both pharmacokinetics and pharmacodynamics [152] (Table 1). This table compare different properties of IL-2, IL-15 and IL-21 and their application described in some detail further.

Table 1. Comparison of cytokines in cancer immunotherapy, highlighting mechanisms, benefits, and limitations.

Cytokine

Mechanism of Action

Advantages

Disadvantages

IL-2

Activates T cells and NK cells [37,38, 41,42]

High efficacy; strong synergy with checkpoint inhibitors; engineered forms reduce toxicity [101, 153-154]

Severe toxicities (e.g., VLS); short half-life [43-51]

Activation of Treg [55-59]

IL-15

Promotes survival and proliferation of NK and memory T cells [91-92]

Reduced toxicity compared to IL-2; supports long-term immune memory [155-158]

Limited clinical experience; challenges with large-scale production [155-172]

IL-21

Stimulates B cell differentiation, regulates T cell function, limits Treg differentiation [173-177]

Promotes anti-tumor immunity; potential synergy with checkpoint inhibitors; enhances adoptive T cell therapy [173-177]

Clinical development in early stages; limited trials and indications; slower progression compared to IL-2 [173-177]

To improve IL-2's pharmacokinetic profile, strategies have focused on extending its circulation half-life. This is achieved by attaching the cytokine to molecules like the Fc domain of immunoglobulins or polyethylene glycol (PEG), or by chimerizing IL-2 with antibodies targeting the TME. These modifications help improve its stability and efficacy. In terms of pharmacodynamics, modifications reduce IL-2's binding to the high-affinity IL-2 receptor (IL-2Rα), which is predominantly expressed on regulatory T (Treg) cells. This adjustment allows IL-2 to preferentially stimulate NK and T cells, which are critical for anti-tumor immunity, while limiting its ability to expand Tregs that can suppress immune responses [101, 153] (Table 1).

Engineered IL-2 variants, such as NKTR-214, modify IL-2 with PEG molecules to create a longer-acting, inactive form that eventually activates and interacts with the medium-affinity IL-2 receptor. Clinical trials combining NKTR-214 with immune checkpoint inhibitors, including nivolumab and atezolizumab, have shown promising results in melanoma, RCC, and non-small cell lung cancer (NSCLC). While the combination therapy has been well-tolerated, further randomized studies are planned to confirm its benefits over monotherapy [153, 154].

Another approach to improving IL-2’s therapeutic profile involves engineering mutated variants with reduced affinity for IL-2Rα, such as cergutuzumab amunaleukin, which fuses a mutated IL-2 with an antibody targeting carcinoembryonic antigen (CEA). These fusion proteins are being tested in clinical trials with various therapeutic agents, demonstrating potential in combination treatments with drugs like trastuzumab and atezolizumab [101].

IL-15, another important cytokine, has shown considerable promise in cancer immunotherapy due to its role in supporting NK and CD8+ T cell proliferation without stimulating Tregs. Unlike IL-2, IL-15 does not bind to IL-2Rα, thus avoiding Treg activation and its associated immunosuppressive effects. Initial clinical trials with recombinant IL-15 demonstrated the expansion of NK and CD8+ T cells in patients with advanced melanoma and RCC. However, severe adverse effects such as fever and hypotension led to the halting of some trials at lower doses [155-158] (Table 1).

One major issue with IL-15 is its reliance on IL-15Rα for signaling, which compromises its stability. To address this, several engineered IL-15 variants have been developed, such as hetIL-15 and hetIL-15Fc. These modifications aim to improve its half-life and stability, enabling prolonged NK and T cell expansion. hetIL-15, based on the natural heterodimeric state of IL-15 and IL-15Rα, has shown promising preclinical results and is undergoing clinical trials for treating metastatic and unresectable solid tumors. hetIL-15Fc, a glycosylated version of IL-15 fused with the Fc region of human IgG1, also demonstrates enhanced efficacy in murine models, supporting the benefits of improved stability for sustained immune activation [159-163].

N-803, another engineered IL-15 superagonist, combines IL-15 with an IL-15Rα sushi domain and an Fc fragment, resulting in a remarkable half-life and bioactivity. Preclinical studies suggest that N-803 enhances NK cell cytotoxicity and can eliminate established tumors. Clinical trials have confirmed its tolerability and efficacy, positioning N-803 as a promising candidate for advanced cancer treatments [164-166].

ALT-803, another engineered version of IL-15, fuses IL-15 with the IL-15Rα sushi domain and an IgG1 Fc domain, improving its half-life and anti-tumor effects. Clinical trials of ALT-803 have shown increased NK and CD8+ T cell expansion in patients with hematological cancers and metastatic non-small cell lung cancer (NSCLC), with good tolerability, especially in combination with PD-1 inhibitors. Its promising results and fast-track designation from the FDA for bladder cancer treatment highlight IL-15's potential [167-170].

Engineered IL-15 therapies offer several advantages over IL-2, primarily in reducing toxicity while enhancing immune cell expansion. Unlike IL-2, which stimulates both effector immune cells and Tregs, IL-15 preferentially expands NK and memory T cells, leading to more selective anti-tumor activity. IL-15 has also been successfully incorporated into adoptive cell therapies, including CAR-T cells, where it improves cell expansion, persistence, and efficacy. Additionally, IL-15 is being tested in combination with oncolytic viruses and tumor-conditional IL-15 pro-cytokines to induce localized immune expansion with minimal toxicity. Despite these advancements, safety concerns remain, requiring careful evaluation in clinical trials [155, 171, 172] (Table 1).

IL-21, a member of the IL-2 family, is another cytokine under investigation in cancer immunotherapy. Like IL-2, IL-21 shares the common gamma receptor (γc) with IL-2 and IL-15, influencing the activation and proliferation of immune cells, including T cells and B cells. IL-21 has garnered attention for its ability to stimulate B cell differentiation into plasma cells, enhance immunoglobulin production, regulate CD4+ and CD8+ T cell responses, and limit Treg differentiation. This makes it a promising candidate for enhancing anti-tumor immunity, similar to IL-2’s effects on T and NK cells [173] (Table 1).

However, despite these similarities, the clinical development of IL-21 as an anti-cancer therapy has progressed more slowly than IL-2. IL-2, with its proven success in treatments for melanoma and renal cell carcinoma, has seen extensive use in cancer immunotherapy and significant clinical trials. In contrast, IL-21’s clinical trials are still in early stages. While IL-21 has been tested alone and in combination with therapies such as ipilimumab, nivolumab, sunitinib, rituximab, sorafenib, and doxorubicin (e.g., NCT00095108, NCT01489059), its progress remains modest. In a Phase I dose-escalation trial, only 3 out of 26 patients experienced partial responses, although pharmacodynamic effects on tumor immunity were observed. Additionally, when combined with rituximab in patients with non-Hodgkin lymphoma, IL-21 showed clinical responses in 8 out of 19 patients [174-176] (Table 1).

Unlike IL-2, which is already in use for cancer treatment, IL-21’s clinical testing is still in its infancy, with only 14 clinical trials conducted since its development began in 2004. These trials have primarily focused on hematological cancers, melanoma, and renal cell carcinoma. In some cases, IL-21 is being tested as a fusion protein with albumin to improve its pharmacokinetics, though the natural form of IL-21 is more commonly used [177] (Table 1). While IL-2 has established itself as a cornerstone of cancer immunotherapy, the potential of IL-21 is still being explored, particularly in combination with other treatments. Given its effects on both T cells and B cells, as well as its ability to limit Treg differentiation, IL-21 holds promise as a future therapeutic option, especially in combination regimens. However, its development lags behind IL-2, and further research is needed to determine its full clinical potential.

Reviewer 2 Report

Comments and Suggestions for Authors

This is a very nicely written and well-compiled review. I really like to read it.

Author Response

Thank you very much for the positive evaluation of our work.

Reviewer 3 Report

Comments and Suggestions for Authors

A well presented narrative review of the current evidence.

Interesting read is the importance of radiotherapy in achieving an abscopal effect and radio sensitising through an immune system.

Author Response

(The authors gave the same response as above.)
